# Artificial Intelligence in Urologic Robotic Oncologic Surgery: A Narrative Review

**DOI:** 10.3390/cancers16091775

**Published:** 2024-05-04

**Authors:** Themistoklis Bellos, Ioannis Manolitsis, Stamatios Katsimperis, Patrick Juliebø-Jones, Georgios Feretzakis, Iraklis Mitsogiannis, Ioannis Varkarakis, Bhaskar K. Somani, Lazaros Tzelves

**Affiliations:** 12nd Department of Urology, Sismanoglio General Hospital of Athens, 15126 Athens, Greece; bellos.themistoklis@gmail.com (T.B.); giannismanolit@gmail.com (I.M.); stamk1992@gmail.com (S.K.); imitsog@med.uoa.gr (I.M.); medvark3@yahoo.com (I.V.); 2Department of Urology, Haukeland University Hospital, 5009 Bergen, Norway; jonesurology@gmail.com; 3School of Science and Technology, Hellenic Open University, 26335 Patras, Greece; georgios.feretzakis@ac.eap.gr; 4Department of Urology, University of Southampton, Southampton SO16 6YD, UK; bhaskarsomani@yahoo.com

**Keywords:** urologic surgery, machine learning, robotic surgery, artificial intelligence

## Abstract

**Simple Summary:**

Robot-assisted surgery facilitates the examination and improvement of artificial intelligence (AI) integration in surgical processes through the provision of comprehensive telemetry data and an advanced viewing interface. Machine learning (ML) techniques enhance the feedback on the development of surgical abilities, the efficacy of the surgical operation, surgical guiding, and predicted results. By incorporating tension sensors on the robotic arms and employing augmented reality techniques, the surgical experience can be greatly improved. This enables the continuous monitoring of organ movements in real time, resulting in enhanced precision and accuracy. The integration of artificial intelligence (AI) into robotic surgery is anticipated to have a substantial influence on the education of upcoming surgeons and improve the entire surgical process. Both endeavours strive for ultimate accuracy in order to enhance the quality of surgical care.

**Abstract:**

With the rapid increase in computer processing capacity over the past two decades, machine learning techniques have been applied in many sectors of daily life. Machine learning in therapeutic settings is also gaining popularity. We analysed current studies on machine learning in robotic urologic surgery. We searched PubMed/Medline and Google Scholar up to December 2023. Search terms included “urologic surgery”, “artificial intelligence”, “machine learning”, “neural network”, “automation”, and “robotic surgery”. Automatic preoperative imaging, intraoperative anatomy matching, and bleeding prediction has been a major focus. Early artificial intelligence (AI) therapeutic outcomes are promising. Robot-assisted surgery provides precise telemetry data and a cutting-edge viewing console to analyse and improve AI integration in surgery. Machine learning enhances surgical skill feedback, procedure effectiveness, surgical guidance, and postoperative prediction. Tension-sensors on robotic arms and augmented reality can improve surgery. This provides real-time organ motion monitoring, improving precision and accuracy. As datasets develop and electronic health records are used more and more, these technologies will become more effective and useful. AI in robotic surgery is intended to improve surgical training and experience. Both seek precision to improve surgical care. AI in ‘’master–slave’’ robotic surgery offers the detailed, step-by-step examination of autonomous robotic treatments.

## 1. Introduction

There has been a growing trend in the use of minimally invasive robotic surgery for significant uro-oncological procedures [1]. The implementation of this technology has resulted in a substantial revolution in the realm of surgery, signifying a remarkable progression in the pursuit of the most effective and least invasive treatment alternatives for patients. Currently, robotic surgery employs sophisticated platforms that operate on a “master–slave” model. In this configuration, the surgeon operates the robotic arms from a distant site, while situated behind an advanced panel. The collaboration between humans and machines enables precise monitoring of the robot’s actions and the utilisation of this data to enhance the surgeon’s input through the assistance of artificial intelligence (AI) [1].

Enhancing surgical precision through the assistance of robotics requires a heightened level of technological intricacy. In order to attain surgical expertise, one must undergo extensive training and develop advanced models [1]. Cutting-edge technological advancements, such as artificial intelligence (AI), have the potential to greatly transform the educational experience in this specific field. Moreover, these cutting-edge technologies can be utilised as invaluable tools by experienced surgeons, augmenting their understanding of the surgical area [1]. As demonstrated, the utilisation of augmented reality has the potential to significantly improve surgical guidance by seamlessly integrating virtual components into the surgical procedure [1].

This study aims to examine the advancements made by artificial intelligence in the field of urology, specifically in the realm of robotic surgery. This article aims to provide valuable insights for urologists in their clinical practice, allowing them to stay updated and potentially incorporate these findings into their daily work. This article begins by examining the concept of machine learning. It then explores the use of artificial intelligence in the training and evaluation of surgeons. Finally, it discusses how artificial intelligence can address the challenge of haptic feedback deficiency. Subsequently, the article delves into the impact of this phenomenon on the field of logistics and the surgical procedures. Lastly, it delves into the potential of artificial intelligence in predicting postoperative outcomes. The concluding section of this article outlines the potential future paths and constraints of artificial intelligence. 

## 2. Methods

In order to conduct a thorough review, we conducted extensive searches in reputable databases such as PubMed/MEDLINE and Google Scholar. Our search covered a period from 1990 to December of 2023. The study incorporated the use of “urologic surgery” in conjunction with cutting-edge technologies such as “artificial intelligence”, “machine learning”, “neural network”, “automation”, and “robotic surgery”. The terms were reviewed by two authors. Any disagreements were resolved through the involvement of a third author.

Eligible studies were original articles conducted in humans, while animal studies, conference abstracts, and reviews have been excluded. Only studies published in English language were considered eligible for inclusion. 

## 3. Main Text

### 3.1. Machine Learning and Deep Learning in Artificial Intelligence

AI systems are composed of computer systems that have been trained to address problems by emulating human intelligence. Machine learning (ML) and deep learning (DL) are highly specialised domains of study within the realm of artificial intelligence (AI) that empower computers to make accurate predictions through the analysis of intricate data patterns [2]. 

The precise evaluation and analysis of clinical information are crucial for providing individualised patient care. Physicians frequently encounter the task of analysing intricate information to diagnose and treat urologic diseases, given the wealth of comprehensive and reliable data available. Through the utilisation of cutting-edge computational techniques, the optimisation of data mining and interpretation can be achieved, leading to a notable improvement in the quality of patient care [2]. Artificial intelligence (AI) is widely used in many areas of daily life for the quick analysis of complex datasets using advanced algorithms and statistical models [3]. Machine learning (ML) has become a prominent and fascinating area within the field of AI. It involves the development of algorithms that are capable of accurately identifying intricate patterns and making predictions. The precision of these algorithms increases with the addition of more data inputs. The incorporation of machine learning (ML) in urologic surgery has demonstrated significant enhancements in the precision and effectiveness of personalised patient care [4]. Through the analysis of complex and interconnected patient health data, which includes a range of factors and non-linear relationships, machine learning has the potential to significantly enhance the quality of care [4].

Machine learning has found extensive applications in various medical fields, enhancing the accuracy of disease diagnosis, aiding in therapy selection, facilitating patient monitoring, and assisting in risk assessment for primary prevention [3,5]. In order to improve surgical systems, the use of machine learning techniques is essential [4]. The automated analysis of patient imaging and precise tracking of surgical anatomy and instruments in the operating area during the perioperative period are crucial aspects to consider [4]. While there is currently no surgical system capable of performing these operations independently, a number of robots have demonstrated encouraging outcomes in tasks such as anatomic tracking, suturing, and biopsy sampling [6]. In the realm of urology, there has been a rise in the adoption of semi-autonomous surgical systems, such as Aquablation™. Previous research has demonstrated the efficacy of incorporating robotic assistance in therapeutic procedures [7]. Advancements in surgical candidate selection and the emergence of automated surgical robotic systems have the potential to greatly improve surgical precision and enhance patient outcomes. Throughout history, the field of urology has consistently been a pioneer in embracing cutting-edge technology, setting itself apart from other surgical specialties. However, despite the progress made in enhancing the safety, efficiency, and design of robotic technology in urology, the integration of automation has not had a substantial impact on urological practice or other surgical specialties.

Developing precise machine learning algorithms is of utmost importance in order to enhance the evaluation and treatment of urologic conditions [4]. Within the field of autonomous surgery, progress can be achieved through enhancements in machine learning algorithms and their integration into current surgical systems. This novel approach showcases promising possibilities for future advancements, although no current systems of this nature exist.

Machine learning (ML) is a field that utilises artificial intelligence methods to create computational systems capable of emulating cognitive processes (Figure 1). Machine learning has the capacity to perform complex tasks, such as logical reasoning, knowledge acquisition, and problem-solving, using advanced nonlinear modelling techniques. Specifically, it allows for the creation of computer algorithms that are not programmed with predetermined rules. Through exposure to sample data, the computational algorithm is able to identify and adjust to distinct patterns within the dataset. Consequently, this technique can be employed to analyse unfamiliar data [8]. Machine learning systems have demonstrated immense value in various domains, showcasing their ability to perform diverse tasks such as predicting traffic patterns, identifying spam, generating text, and enhancing online marketing strategies [8]. Furthermore, the capacity to create non-linear algorithms is of great importance in the analysis of medical data, which frequently exhibits complex patterns and subtle details [8]. Accessing reliable and comprehensive datasets is crucial for effectively training any machine learning system [8]. Through the utilisation of extensive datasets, sophisticated algorithms can be trained to effectively process inputs with inconsistencies and provide precise outcomes [8]. Similarly to statistical models, comprehensive retrospective datasets may sometimes have missing data points, confounding factors, and biases that may impact the training of algorithms. Using a validated dataset enables the assessment of method accuracy [8].

Machine learning encompasses a broad array of subfields that are dedicated to the analysis and understanding of various forms of data [10]. Research in various subfields has shown that machines have the ability to operate and make decisions without relying on explicit executable instructions [10]. Instead, individuals can acquire knowledge and make predictions by recognising patterns and drawing conclusions [8,10]. For instance, the field of natural language processing (NLP) is dedicated to the study of how computers can effectively analyse and understand human language [10]. Computer vision (CV) encompasses the automated analysis of images or videos, including radiographic or endoscopic images [10]. In the field of medicine, a significant amount of research has been dedicated to the study of artificial neural networks (ANN) [10]. The ANN is composed of processing nodes that bear resemblance to biological neurons, organised in a layered structure [10]. This architecture enables computer systems to acquire knowledge and discern patterns from intricate inputs [10]. The layers are composed of an input layer containing input nodes, a hidden layer containing hidden nodes, and an output layer containing output nodes [8]. The properties of these layers, with regard to their depth and width, play a crucial role in determining the functionality of the artificial neural network (ANN) [10] (Figure 1). Expanding the dimensions of a system can improve its processing and learning abilities [10].

The utilisation of advanced computational tools can greatly assist in analysing complex medical data, leading to improvements in clinical and surgical procedures. The integration of different subfields of machine learning is essential for the advancement of autonomous surgical systems. This facilitates the examination and integration of various forms of data. The potential of these technologies is found in their capacity to assist in image interpretation, monitor and strategize operations, administer medical treatments, and forecast outcomes.

### 3.2. Evaluating Surgical Expertise

In light of the rising public apprehension regarding the skill level of surgeons, along with the surge in robotic surgery utilisation, there is an amplified demand for efficient training models and dependable assessment tools to evaluate clinical competency [1]. Different strategies have been developed to improve traditional training methods, as advancements in surgical techniques require specialised training in both technical and non-technical skills [1].

In a study by Fard et al. the competence of robotic surgery abilities was objectively evaluated using three distinct machine learning algorithms, which were chosen for their suitability in analysing complex surgical performance data [11]. The methods employed included k-nearest neighbours (k-NN), logistic regression, and support vector machines (SVMs) [11]. Each method offers unique advantages and potential drawbacks. k-NN is intuitive and effective, particularly when there is no clear assumption about the underlying data distribution, but it can be computationally intensive and sensitive to the scale of the dataset and the choice of k. Logistic regression, while providing a probabilistic output and being less prone to overfitting through regularisation, assumes a linear relationship between the independent variables and the log odds, potentially underperforming with non-linear decision boundaries [11]. SVM is highly effective in high-dimensional spaces and in cases where there is a clear margin of separation; however, it requires careful tuning of hyperparameters and kernel choice and can become computationally demanding with large datasets [11]. In their findings, Fard et al. achieved high classification accuracies, with SVM performing best in scenarios demanding robustness against overfitting, particularly with nonlinear data patterns [11]. The classification accuracies reached up to 89.9% for certain surgical tasks, highlighting the potential of these machine learning approaches to significantly enhance the objective assessment of surgical skills based on quantitative analysis of movement trajectories during surgery [11]. According to the data, the authors have successfully shown a significant level of precision in the automated ML algorithm’s capacity to distinguish between novice and experienced surgeons [11]. This differentiation can be established in a matter of seconds upon finishing tasks [11]. However, in order to ensure the accuracy of this model on a larger sample size, further research is required due to the limited number of participants [11].

Extensive research has been conducted on the application of machine learning in the monitoring and analysis of surgical techniques. As an illustration, Ghani et al. analysed recordings of bladder neck anastomosis performed by 11 surgeons [12]. The aim of their study was to provide comprehensive and step-by-step education to a computer vision algorithm on recognising the speed, trajectory, and smoothness of instrument movement, as well as the instrument’s positioning in relation to the opposite side [12]. The surgeons were later classified according to their level of expertise, distinguishing between those with exceptional talent and those with limited skill [12]. The system underwent several validations using a final video, and the results were averaged [12]. The evaluations were compared to a blinded review conducted by 25 peer surgeons using the global evaluative assessment of robotic abilities (GEARS) instrument [12]. The programme demonstrated a remarkable accuracy rate of 83% in skill categorisation when analysing individual instrument points, which rose to 92% when considering joint movement as well. By incorporating the contralateral instrument, the categorisation of binary skill levels was significantly improved to a remarkable 100% accuracy [12]. According to the study, the researchers discovered a strong correlation between specific parameters and expertise [12]. The study examined the correlation between needle driver forceps and joint position, along with variables such as acceleration and velocity [12].

Wang et al. further advanced their research by creating a sophisticated deep learning model that incorporates an artificial neural network, a highly intricate machine learning technique [13]. This model is based on the inter-neuronal connections observed in biological nervous systems. It employs a variety of machine learning techniques to effectively analyse data and extract insights from incoming information [13]. The primary utilisation of DL algorithms was observed to be pattern recognition [13]. Consequently, their approach was employed to ascertain the motion characteristics during the training of robotic surgery [13]. A total of eight surgeons underwent evaluation in a series of five trials involving various activities, including suturing, knot-tying, and needle passage [13]. The precision rate of the artificial neural network model was found to be over 90% in evaluating the proficiency of robotic surgical skills. The evaluation was conducted within a time frame of 1 to 3 s, as stated in the study [13]. This feedback was given without the requirement of segmenting gestures or observing the entire trial [13].

A novel method for evaluating machine learning skills was introduced by Ershad et al. in a recent study [14]. The researchers conducted a thorough analysis of the surgeon’s “movement style” in order to evaluate their performance [14]. One key factor in this approach is the recognition that a skilled surgeon can execute manoeuvres with expertise and effectiveness, displaying a composed and coordinated demeanour, which sets them apart from less experienced surgeons [14]. Information regarding surgical movements was collected from a cohort of 14 surgeons with varying degrees of proficiency in robotic surgery [14]. A framework based on sparse coding was utilised to automatically identify stylistic behaviour within short time intervals [14]. This framework is built upon the acquisition of position data from various parts of the upper limb, including the hands, wrist, elbow, and shoulder [14]. A codebook was generated for each stylistic term by utilising the positive and negative labels obtained for each trial through crowd sourcing [14]. Each surgeon performed two procedures (ring and rail and suture) three times [14]. The movement was tracked using state-of-the-art three-dimensional electromagnetic technology, which allowed for the precise monitoring of the surgeon’s shoulder, wrist, and hand positions during virtual simulator training [14]. The classification of instructional videos was accomplished through crowdsourcing, employing various adjectives to characterise the behavioural approach [14]. Subsequently, this was employed to train a classifier model [14]. According to the authors, there was a significant improvement of 68.5% in the accuracy of skill level classification when compared to the raw kinematic data collection [14]. One benefit of this technique is that it eliminates the need for extensive surgical expertise when evaluating surgical proficiency [14]. Instead, it considers the qualitative aspects of motion that are specific to the surgeon, such as smoothness, calmness, and coordination, rather than solely concentrating on the task at hand [14].

A study conducted by Youssef et al. introduced an algorithm that integrated videolabelling into 25 robot-assisted radical prostatectomy (RARP) surgeries [15]. The videos were integrated into Proximie, an augmented reality platform, and provided to a novice surgeon for the purpose of annotating the 12 steps of the procedure [15]. Of the 25 films analysed, 17 were deemed appropriate for categorisation, while 8 were excluded due to their subpar quality and time-consuming nature [15]. The accuracy of temporal tagging ranged from 85.6% to 100%, with an average accuracy of 93.1% [15]. Studies used in evaluating surgical expertise are listed in Table 1.

### 3.3. Addressing the Issue of Haptic Feedback

Robotic surgery utilises advanced 3D visualisation to enhance the surgeon’s skill and precision. Nevertheless, the actions required for dissection, the amount of pressure exerted, and the assessment of tissue responses are all reliant on visual cues. In the absence of haptic feedback, the results of surgical procedures may be adversely affected. This is due to the potential danger of exerting excessive force on delicate tissues, disrupting anatomical structures, and compromising the integrity of sutures. Excessive force used during robotic radical prostatectomy (RARP) can have detrimental effects on the neuro-vascular bundles, resulting in neuropraxia and potentially causing a delay in the recovery of sexual function [1]. On the other hand, if the force applied during the knot-tying process is not enough, it may lead to poor suture retention.

In a recent study, the researchers Dai et al. successfully developed an advanced warning system designed to detect suture breakage [16]. A novel system was developed and validated by the authors, which integrates a grasper with biaxial shear detection and haptic feedback [16]. The purpose of this system is to notify the operator in advance of a potential suture rupture [16]. In addition, the design enables the convenient manipulation of sutures without compromising their effectiveness, as assessed by evaluating the tightness of the resulting suture knots [16]. The Da Vinci robotic surgical system was fitted with biaxial shear sensors [16]. This system offers vibrotactile feedback as the tension on the suture nears its maximum limit [16]. A recent study has unveiled notable advancements in surgical outcomes following the implementation of a haptic feedback system [16]. The results revealed a significant 59% decrease in suture breakage, a 3.8% reduction in knot slippage, and improved consistency in repeated tasks for inexperienced surgeons [16]. In addition, the surgeons were able to achieve significantly lower rates of suture rupture. The initial four knots had a rate of 17%, while the subsequent six knots had a rate of 2% (*p* 0.05). This improvement was attributed to the utilisation of the tactile feedback system during practice [16]. However, the absence of haptic feedback during surgical training resulted in a significant occurrence of suture breakdown, as reported in a study [17].

Piana et al. used three-dimensional augmented reality guidance during kidney transplantation (KT), which could help surgeons navigate the surgery, enhancing safety for patients with atheromatic vascular disease. This technology removes the need for haptic input, which is not present in robotic consoles. Utilising high-accuracy computed tomography scan imaging, three-dimensional virtual models were generated. The models were subsequently overlaid onto the vasculature in the context of robot-assisted kidney transplantation (RAKT) using the Da Vinci console software program [18].

An assessment was conducted to determine the correlation between virtual models and the real anatomical structures of patients. This involved a comparison of measurements pertaining to arteries and plaques. The assessment centred on the structure of the iliac plaques and the precision of the virtual models in representing them. The assessment of the virtual model’s effectiveness in overlaying the actual blood arteries involved examining the presence of plaques in individuals who underwent RAKT with living donor grafts. One of the main limitations lies in the need for proper training to ensure precise alignment of virtual models with the real environment. The implementation of three-dimensional augmented reality (3D AR) allowed surgeons to address a key limitation of RAKT, paving the way for expanding its potential uses to individuals with advanced atheromatic vascular disease [18]. Studies used in haptic feedback research are listed in Table 2.

### 3.4. Exploring the Intersection of Artificial Intelligence and Logistics

Zhao et al. revealed that the accuracy of time estimation for robotic procedures can be greatly improved by employing a machine learning approach [19]. A recent study conducted by researchers has resulted in the development of a predictive model for estimating the duration of robotic surgeries [19]. The researchers took into account a range of factors, such as patient characteristics such as age, obesity, malignancy, tumour location, and comorbidities [19]. The researchers also considered factors such as the type of procedure, the specific model of the robotic system used, and the level of expertise of the tableside assistant, which varied from resident to attending [19]. A machine learning model was created by analysing data from 424 surgical procedures in the areas of robotic urology, gynaecology, and general surgery [19]. The implementation of this predictive model resulted in a notable enhancement in the accuracy of predicting OT, with a 16.8% increase [19]. This finding indicates that utilising the model has the potential to improve the daily schedule of operative time, resulting in a significant optimisation of operating room resources [19]. Studies used in surgical logistics are listed in Table 3.

### 3.5. Preoperative Identification of Exact Anatomy

To improve surgical procedures, the first step is to create programmes that can consistently and accurately identify patient anatomy. In the field of surgical treatment, the interpretation of radiologic imaging holds great significance. As a result, there have been numerous endeavours to integrate machine learning techniques into the analysis of imaging.

Computer vision (CV) is a subfield of machine learning that focuses on analysing images. It has the potential to improve the identification and diagnosis of urologic conditions. For example, in the realm of stone disease, several studies have demonstrated that employing cardiovascular algorithms on CT abdominal imaging data can precisely identify the precise location of the stones [20]. The effectiveness of these solutions is based on the advanced processing of image signals, which allows for the development of more robust algorithms [21]. By employing this technique, algorithms are able to precisely identify even the most minute visual distinctions between abnormal and healthy anatomical structures [21].

In addition, through the use of segmentation techniques, computational algorithms can autonomously detect and delineate the precise surgical anatomy of interest by utilising anatomical localisation in the imaging data. Presently, this procedure is conducted through manual means and entails the utilisation of various surgical techniques that employ image guidance. However, the procedure of physically isolating the desired anatomy and blood vessels can be quite arduous and time-consuming. Automating this process necessitates the utilisation of computer vision algorithms to swiftly analyse medical images and precisely interpret anatomical details. Recent studies have shown that the use of multi-atlas segmentation has resulted in better identification of anatomical variations and increased accuracy in segmenting structures [22]. This method entails examining a patient’s anatomy through the utilisation of an algorithm that has been extensively trained on a large dataset of other imaging information [22]. This technique has been extensively studied to aid in the process of surgical planning [22]. In two recent studies, multi-atlas segmentation was employed to examine the pyelocaliceal anatomy of patients [23,24]. A CT urography technique was employed to automatically analyse the 3D anatomy, obviating the need for manual measurement assessment. The segmentation process was effectively performed on 8 out of 11 pyelocaliceal systems to quantify the infundibulopelvic angle (IPA) [23,24]. In the image labelling process, a few errors were identified. However, the work effectively demonstrates the potential of utilising multi-atlas segmentation for anatomical characterisation [23,24].

Ongoing progress in the methodology will facilitate the production of accurate anatomical features and improve the depiction of complex patient anatomy that is difficult to capture using current techniques [25].

Precise monitoring of the patient’s anatomy during surgery requires careful consideration of various factors that can potentially obstruct the visual field. These factors include equipment, blood, smoke, and adipose tissue. Furthermore, it is crucial to be able to anticipate the manner in which the tissues will undergo deformation throughout the dissection procedure. Despite the challenges, researchers are making significant progress in improving the automated recognition of patient anatomy. A recent study by Nosrati et al. introduced a new technique for synchronising pre-operative data with intraoperative endoscopic images in the context of partial nephrectomy [26]. The alignment technique employed subsurface feature cues, including vessel pulsation patterns, in conjunction with colour and texture data, to autonomously register the workspace with the preoperative imaging [26]. The examination of colour and textural visual cues is conducted through the utilisation of machine learning techniques [26]. This study showcases a groundbreaking endeavour in utilising vascular pulsation cues to facilitate the preoperative to intraoperative registration process [26]. In addition, their framework incorporates a deformation model that is customised for specific tissues, enabling the authors to effectively manage the non-rigid deformation of structures that occurs during surgery [26]. A study conducted by Di Dio et al. employed computer vision techniques to develop an exceptionally precise and thorough three-dimensional representation of a kidney, utilising advanced artificial intelligence algorithms [27]. Subsequently, this model was utilised to perform a partial nephrectomy on a tumour measuring 35 × 25 cm situated on the anterior aspect of the upper renal region [27]. Utilising this method, the surgeon intentionally exerted pressure on the artery responsible for supplying blood to the tumour and conducted the surgical procedure [27]. A recent study conducted by Klen et al. focused on the development of a machine learning model that aimed to analyse preoperative risk factors linked to postoperative mortality following radical cystectomy [28]. A total of 1099 patients who underwent radical cystectomy (RC) surgery at 16 hospitals in Finland from 2005 to 2014 were included in the dataset [28]. The model achieved an area under the curve (AUC) value of a 0.73 [28]. Several notable risk factors have been identified, including the American Society of Anesthesiologists (ASA) physical status classification, congestive heart failure (CHF), age-adjusted Charlson comorbidity index (ACCI), and chronic pulmonary conditions [28]. In accordance with previous research findings [25], physicians can provide valuable guidance during surgical consultations by identifying patients who may be more prone to experiencing complications from the procedure [28]. A recent study by Checcucci et al. introduced a model that performed a comparative analysis of prostates using three-dimensional (3D) models. The study compared patients who had undergone a robot-assisted radical prostatectomy (RARP) procedure with those who had traditional non-3D models [29]. An assessment was made on positive surgical margins (PSM) using multivariable linear regression (MLR) models [29]. Based on the research results, individuals who utilised 3D models experienced a reduced incidence of positive surgical margins in comparison to those who did not (25% vs. 35.1%, *p* = 0.01) [29]. According to MLR models, the presence of a 3D model and the absence of extracapsular extension on mpMRI were identified as independent factors that significantly decreased the chances of positive surgical margins (PSM) [29]. Studies with preoperative applications are listed in Table 4.

### 3.6. Intraoperative Application of Artificial Intelligence in Surgical Procedures

In order to accomplish real-time, automated, intraoperative interventions, it is essential to create a sophisticated artificial intelligence platform that efficiently utilises machine learning. The platform must possess the capability to precisely detect patient anatomy and monitor the equipment utilised, all the while adjusting to the dynamic conditions of the surgical setting. This device has the potential to significantly assist doctors in making crucial decisions during surgical procedures or provide instant feedback on surgical techniques. Several studies have established a foundation for the future development of automated machine learning (ML) systems designed for use in the operating room. A recent study conducted by Baghdadi and colleagues utilised machine learning techniques to analyse colour and texture in order to identify anatomical components during pelvic lymph node dissection [30]. The study sought to make predictions regarding the quality of the dissection based on these findings [30]. The findings of the automated skills assessment model demonstrate a significant association with expert evaluations of lymph node dissection quality, achieving an accuracy rate of 83.3% [30]. This presents an opportunity for further evaluation of these training tools [30]. A study conducted by Haifler et al. demonstrated the potential of machine learning analysis of short-wave Ramen spectroscopy data in accurately distinguishing between renal cell cancer and benign renal tissues [31]. The study was carried out in a controlled laboratory environment using a specialised workstation [31]. The accuracy, sensitivity, and specificity achieved were 92.5%, 95.8%, and 88.8%, respectively [31]. This technique enables the immediate evaluation of surgical margins during procedures, regardless of lighting conditions [31].

Augmented reality (AR) holds promise in providing live assistance during surgical procedures and could be beneficial in ensuring secure navigation on upcoming autonomous robotic-assisted surgery (RAS) platforms. A recent study by Checcucci et al. introduced a highly advanced artificial neural network model for predicting occurrences of bleeding during robotic prostatectomy surgeries [32]. The programme analyses the footage captured by the endoscope at regular intervals of 3 s and provides a confidence rating [29]. Confidence ratings below 100% may indicate a potential risk of bleeding [32]. The model showcases a remarkable true positive rate of 98% [32].

A study conducted by Porpiglia et al. utilised MRI to generate intricate prostate models [30,31]. Surgeons were able to conduct a thorough examination of cancer characteristics during prostatectomy, particularly in cases where the cancer had extended beyond the prostate capsule, also referred to as extracapsular extension or ECE [33,34]. In recent advancements, researchers have made significant progress in integrating 3D models into live surgeries through the da Vinci surgical console image. Building upon this success, they have now devised a computer vision algorithm to enhance the alignment of virtual 3D models with the real-time surgical view of the prostate [35]. As per established models, metallic clips were strategically positioned on suspected areas of extracapsular extension (ECE) prior to the removal of the neurovascular bundle [35]. The conclusive pathological examination confirmed the existence of cancer in all examined regions among patients with pT3 stage [32]. In a recent study, researchers conducted a comparison between surgical teams utilising 3D AR guidance and those who did not [34]. The findings were highly notable, as the implementation of the virtual reality model greatly improved the identification of extracapsular extension (ECE) [34]. There was a significant improvement, as the rate of detection increased from 47.0% to a flawless 100% (*p* < 0.002) [34]. The authors highlight the wide range of potential applications for this technique, extending beyond prostate surgery to include robotic partial nephrectomy [34]. This is particularly significant for tumours that are endophytic or situated towards the posterior region [34]. Additional validation is necessary, but the use of 3D augmented reality (AR) has the potential to enhance intraoperative navigation and optimise the delicate balance between minimising positive surgical margins and maximising functional preservation [34].

In their study, DeBacker et al. employed augmented reality technology in the realm of renal surgery [36]. A cutting-edge deep learning algorithm has been created to precisely detect and identify different inorganic substances, such as surgical equipment, in real time during surgical procedures [36]. The system has successfully acquired the ability to extract valuable information from a dataset consisting of 65,927 meticulously annotated instruments, as performed by human experts [36]. The instruments are distributed among 15,100 frames, offering a wide array of data [36]. Nevertheless, the author did come across certain constraints with the model [36]. Alignment issues and the presence of motion artefacts during the subject’s breathing were observed [36].

The effectiveness of augmented reality robot-assisted partial nephrectomy (AR-RAPN) is hindered by the requirement for continuous manual alignment of the highly precise 3D virtual models with the actual anatomy. In a study by Amparore et al., the authors presented their initial findings on the implementation of an automatic 3D virtual model overlapping during AR-RAPN. In order to achieve a completely automated HA3D model overlapping, their approach involved the implementation of computer vision techniques that relied on the recognition of specific points of reference to establish connections within the virtual model. Given the restricted visual scope of RAPN, the entire kidney was utilised as a reference point. In addition, in order to address the challenge of colour similarity between the kidney and its surrounding structures, the researchers utilised the NIRF Firefly fluorescence imaging technology to greatly enhance the visibility of the organ. A software called “IGNITE” was developed for the purpose of automatically anchoring the HA3D model to the real organ. This software takes advantage of the enhanced view provided by the NIRF technology. A total of ten automated AR-RAPN procedures were conducted. An HA3D model was created and displayed as an augmented reality image within the robotic console for all patients. Throughout the surgical procedures, the automatic ICG-guided AR technology effectively connected the virtual model to the actual organ without any manual assistance (average connection time: 7 s). This was achieved even when the camera was moved around the operative field, and when the organ was zoomed in or moved. Automatic AR technology successfully identified renal masses in seven patients with totally endophytic or posterior lesions, allowing for a successful enucleoresection. There were no complications of Clavien >2 during or after the surgery, and no positive surgical margins were observed [37].

The same author has published a prospective study that examined two different approaches for automatically overlaying a model onto the actual kidney. The first method involved utilising computer vision technology, which utilised the enhanced images of the kidney obtained through intraoperative indocyanine green injection. The second method involved using convolutional neural network technology, which involved training the network on frames from prerecorded surgery videos and then processing live endoscope images. A team of experts in bioengineering, software development, and surgery collaborated to develop highly precise 3D models for autonomous 3D-AR-guided RAPN. Demographic and clinical information was collected for each participant. Two separate groups were established for the evaluation of different technologies. Group A consisted of 12 patients, while group B included 8 patients. The preoperative and postoperative characteristics exhibited similarities. The mean co-registration time for the initial technology was 7 (3–11) seconds, while the second technology had an average time of 11 (6–13) seconds. There were no significant complications during or after the surgery. The functional outcomes exhibited no significant differences between the groups at any of the recorded time points. The 3D model was securely attached to the kidney with minimal manual adjustments. A novel method was employed to enhance the detection of kidneys without the need for indocyanine injection, resulting in improved recognition of organ boundaries during tests [38].

Augmented reality (AR) may automatically identify significant anatomical features and integrate preoperative imaging, such as prostate multiparametric MRI. The findings of this study hold promising implications for the future development of autonomous soft-tissue surgery. Nevertheless, the progress of this technology is currently in its nascent phase and will require a considerable amount of time before it can be incorporated into forthcoming autonomous RAS platforms. Studies with postoperative applications are listed in Table 5.

### 3.7. Postoperative Outcomes Prediction in Surgical Procedures

A recent study conducted by Chen et al. has demonstrated the superior effectiveness of AI systems trained on clinical, pathologic, imaging, and genomic data in predicting treatment outcomes compared to the traditional D’Amico risk stratification [39]. This discovery signifies a noteworthy progress in tailoring patient care based on their individual needs. Aside from patient-related factors, there are also factors related to the surgeon that can have an impact on the outcome of patients after surgery [39].

Furthermore, the information gathered during the surgical procedure can be linked to the patient’s overall clinical results. In a recent study, Hung et al. conducted research to explore the potential of algorithms in predicting surgical outcomes for a specific procedure [40]. In their study, Hung et al. conducted an investigation into automated performance indicators obtained from the data of the “dVLogger”, a recording device connected to the robotic system used in 78 full-length robot-assisted radical prostatectomy (RARP) procedures [40]. The dVLogger is a device that captures video and movement data, and it is manufactured by Intuitive Surgical [40]. Three machine learning algorithms were utilised in this study, using data from robot system and hospital length of stay [40]. The analysis encompassed 25 distinct factors pertaining to the kinematic data of both the overall/dominant and non-dominant instruments [40]. The study examined various factors such as travel time, path length, movement, velocity, and system events such as frequency of clutch use, camera movement, third arm, and energy use [40]. The aim was to assess the accuracy of these features in predicting surgical outcomes [40]. The algorithms underwent training using training material and training labels, specifically focusing on cases with a length of stay of 2 days or less, as well as cases with a length of stay exceeding 2 days [40]. The algorithm with the highest performance was selected. The algorithm utilised in this study classified the cases into two categories: “Predicted as expected LOS (pExp-LOS)” and “Predicted as extended LOS (pExt-LOS)” [40]. A machine learning model achieved a remarkable accuracy rate of 87.2% in predicting the duration of hospital stays [40]. They successfully predicted a hospital stay of over 2 days with an 87% accuracy rate [40]. In addition, a strong correlation was found between the expected patient outcomes, specifically the duration of OT and Foley catheter usage, and the actual results (r = 0.73, *p* < 0.001 for OT; r = 0.45, *p* < 0.001 for Foley catheter duration) [40]. The manipulation of the camera during surgery has been shown to have a significant impact on surgical performance measures [40]. Factors such as the frequency of adjustments, the amount of idle time, and the positioning of the camera all play a crucial role in this regard. Manipulating the camera during robotic surgery has emerged as a promising method for assessing a surgeon’s level of expertise [40]. This study aims to enhance logistics by potentially enabling cost-effective personalised catheter removal timing [40].

In a separate research study, the researchers showcased the effectiveness of their application of automated performance indicators and deep learning models in predicting urine continence (UC) in a group of 100 patients who had undergone robot-assisted radical prostatectomy (RARP) [41]. Specifically, data were collected from 100 RARPs to gather measures of robotic surgical automated performance (APMs), along with patient clinicopathological and continence data [41]. Utilising a DL model known as DeepSurv, they were able to predict the extent of urine continence following surgery [41]. The significance of model features in prediction was assessed through a ranking process [41]. The eight surgeons were categorised into distinct groups, taking into account their top five highest rated attributes [41]. The top four surgeons were categorised as ‘Group 1/APMs’, while the remaining four were classified as ‘Group 2/APMs’ [41]. Another cohort of RARPs performed by these two surgical teams was later used for comparison [41]. Upon analysing the relationship between kinematic data and clinical patient features, a noteworthy finding emerged. The researchers observed that the most precise estimation of UC recovery following RARP typically took place at an average of 4 months [41]. A study conducted to evaluate the precision of this method was compared to solely relying on clinical features [41]. By combining automated performance metrics with clinical features, a concordance index of 59.9% was achieved. However, when relying solely on automated performance metrics, the concordance index dropped to 57.7% [41]. A study reported a concordance index of 56.2% when only clinical features were considered [41]. In this dataset, the performance measures used to evaluate apical dissection and vesicourethral anastomosis were identified as the most significant [41]. In addition, patients who received surgery from surgeons who used precise automated performance measurements experienced a notable increase in continence rate [41]. After a period of 3 months, the patients observed a notable increase of 10.8% in their continence rate. This positive trend continued, with a further increase of 9.1% after 6 months [41]. In contrast, patients who underwent surgery performed by surgeons with less favourable metrics did not experience a similar degree of improvement [41]. The following strategies will be utilised to effectively communicate the findings of our system outcomes research and will have an impact on professional practice and policy making. Moreover, the rapid progress in technology is set to have a substantial influence on the authentication of credentials and the delivery of surgical training. Studies associated with postoperative outcomes prediction are listed in Table 6.

## 4. Current Limitations and Implications for Future Research

The effectiveness of machine learning algorithms is intricately linked to the quality of the input data [42]. For optimal algorithm training, it is essential to conduct comprehensive data pre-processing [36]. Accurate and reliable solutions tailored to individual patients require extensive and varied datasets [42]. Furthermore, it is crucial for physicians to possess a comprehensive comprehension of the data utilised in algorithm training in order to precisely interpret findings, since not all machine learning technologies are intended to be readily explicable [42]. The study discussed is limited by the scarcity of data and the lack of external validation [42]. Nevertheless, the widespread utilisation of electronic health records (EHRs) offers a hopeful prospect of collecting substantial data on a large scale in the coming years. The incorporation of machine learning methods to improve illness detection and treatment will occur in a gradual manner, similarly to the adoption of previous revolutionary technologies. Human supervision is essential for this procedure, as emphasised in previous studies [17,42]. Further investigation is required to delve into and validate the efficacy of machine learning methods in delivering individualised care for patients undergoing urologic surgery. Nevertheless, the effective application of these techniques in different areas of our everyday existence provides us with confidence that these challenges will ultimately be surmounted. The level of autonomy exhibited by robots is determined by three key factors: the intricacy of the task at hand, the specific operational environment, and the intended interaction between humans and robots.

A recent study by Shademan et al. unveiled the Anastomosis Robot (STAR) system, a revolutionary robot specifically engineered to independently perform end-to-end intestinal anastomosis during open surgery [43]. The machine operates in conjunction with human assistance for retraction. In a recent study, researchers have made significant progress in the field of medical technology. They have enhanced the capabilities of STAR by integrating a state-of-the-art 3D imaging endoscope and a meticulously precise suturing planning technique. It has been shown to have a higher level of effectiveness in terms of the accuracy of suture placement and the requirement for suture readjustment when compared to manual suturing [44].

Before commencing clinical practice, it is essential to establish precise criteria for evaluating the efficacy of autonomous robotic surgery [1]. It is crucial to assess specific factors, including the capacity to manage unforeseen situations, accuracy in surgical procedures, and the ability to consistently reproduce them [1]. By integrating automation into their practices, surgeons can attain outcomes that are more consistent and predictable [1].

Nevertheless, there are ethical and safety considerations that arise in the context of autonomous robotic surgery. Obtaining patient consent is crucial prior to using a surgical robot.

Additionally, it is of utmost importance to assess the particular conditions under which the robot has undergone training and the knowledge it has gained throughout this process to prevent the retention of inappropriate information and ensure the welfare of patients. Adapting to unforeseen circumstances or intricate situations is a significant hurdle in the realm of autonomous surgery. Robotic systems often employ an unstructured approach when faced with an unexpected event. Increasing the variety of training models can improve the effectiveness of autonomous surgical robots in various situations.

Obtaining comprehensive informed consent from patients is a top priority for surgeons prior to any surgical procedure. Will future consent forms incorporate a clause that releases the surgeon from liability for technological malfunctions and subsequent complications that may occur during surgery using an autonomous robotic system? The responsibility for errors in autonomous robotic surgery requires a critical inquiry. Could doctors potentially be held accountable for any adverse events that occur during the autonomous phase of a procedure, along with the manufacturers? This would be akin to the shared responsibility seen in semi-autonomous road automobile accidents. In the absence of a collective sense of duty, the utilisation of cutting-edge autonomous robotic surgical systems may necessitate physicians to depend on technology that they may not possess complete comprehension of, yet are still held accountable for under the law. To ensure the seamless integration of autonomous robotic-assisted surgery (RAS) into routine clinical practice, it is imperative for surgeons to actively engage in collaboration with manufacturers. This collaboration is crucial for establishing well-defined roles, especially in situations where patient safety could be jeopardised by malfunctions in autonomous systems. Furthermore, the increasing utilisation of automation in robotic-assisted surgery (RAS) raises questions regarding the continued appropriateness of the ethical and regulatory parameters that surgeons commonly discuss with their patients.

## 5. Conclusions

Utilising machine learning techniques can aid in facilitating patient-centred surgical treatment and enhancing patient involvement in their decision-making processes. Despite being in its early stages of development, the utilisation of comprehensive electronic health record datasets to enhance algorithms exhibits potential in enhancing their efficacy and possible integration into clinical practice. Machine learning systems have the potential to bring about a significant transformation in the field of urology. By efficiently handling vast quantities of data, these systems can offer substantial advantages to both patients and healthcare systems. Nevertheless, the advancement of more robust models will be essential to facilitate the extensive utilisation of AI in training programmes and ultimately enhance patient results.

## Figures and Tables

**Figure 1 cancers-16-01775-f001:**
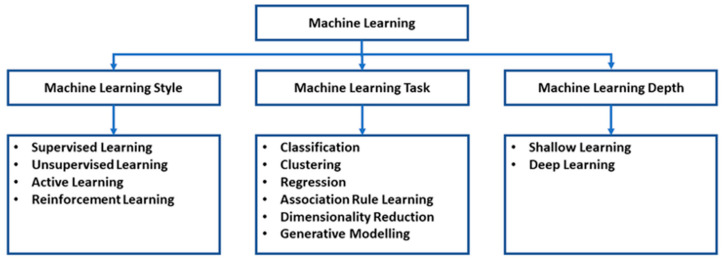
Machine learning classification [9].

**Table 1 cancers-16-01775-t001:** Studies used in evaluating surgical expertise.

Authors ^(Ref)^	Outcome	Data Source	Type of AI	Findings
Fard et al. [11]	Tool for evaluating and measuring skills	Characteristics of the robotic arms on a global scale: duration of task completion, length of the path, perception of depth, velocity, fluidity of movement, curvature angular displacement, tortuosity	Three classification methods: k-nearest neighbours, logistic regression, and support vector machines	Differentiating between skilled and inexperienced robotic surgeons using automated methods
Ghani et al. [12]	Tool for evaluating and measuring skills	Videos of 11 surgeons performing the bladder neck anastomosis, while the algorithm measured the speed, trajectory, and smoothness of instrument movement, along with the instrument’s positioning in relation to the opposite side, in a comprehensive and step-by-step manner	Computer vision algorithm	Certain parameters were closely linked to expertise, like the relationship between needle driver forceps and joint position
Wang et al. [13]	Tool for evaluating and measuring skills	Data regarding the movement of the master tool manipulator and the patient side manipulator	Artificial neural network	Automated evaluation and feedback of surgical proficiency utilising artificial neural network
Ershad et al. [14]	Tool for evaluating and measuring skills	Installed position sensors on the surgeon’s limbs Metrics for measuring stylistic behaviour: liquid/viscous, polished/coarse, sharp/unsteady, rapid/lethargic, tranquil/distressed, calm/anxious indecisive, organised/disorganised	Sparse coding framework	Assessment of surgical expertise through the evaluation of the proficiency in stylistic conduct
Youssef et al. [15]	Tool for evaluating and measuring skills	25 videos of RARP used for videolabelling by a beginner surgeon	Proximie	17 suitable for videolabelling, average tagging accuracy of the steps of 93.1%

**Table 2 cancers-16-01775-t002:** Studies used in haptic feedback research.

Authors ^(Ref)^	Outcome	Data Source	Type of AI	Findings
Dai et al. [16]	Haptic feedback surrogate	Robotic tools equipped with sensors	Not mentioned	A haptic feedback system is designed to enhance the prevention of suture breakage, increase the quality of knots, and facilitate learning
Piana et al. [18]	Haptic feedback surrogate	High-accuracy CT scanning creating 3D images	3D augmented reality guidance during KT, with ML using the Da Vinci console software	Eliminates the need for tactile sensation (haptic feedback) during KT, and facilitates iliac vessels plaques

Abbreviations: ML = machine Learning; KT = kidney transplantation; CT = computed tomography.

**Table 3 cancers-16-01775-t003:** Studies used in surgical logistics.

Authors ^(Ref)^	Outcome	Data Source	Type of AI	Findings
Zhao et al. [19]	Intraoperative outcomes	Variables employed in the construction of the model: prearranged surgical procedure, group of procedures, age, obesity, sex, combined case, robotic prototype, cancer, location of the tumour, hypertension, tobacco usage, atrial fibrillation, obstructive sleep apnoea, coronary artery disease, diabetes mellitus, cirrhosis, chronic obstructive pulmonary disease (COPD), renal insufficiency, ASA classification, calendar month, time of day, anaesthesiologist, operating room aide	ML	Improved precision in forecasting case duration

**Table 4 cancers-16-01775-t004:** Studies with preoperative applications.

Authors ^(Ref)^	Outcome	Data Source	Type of AI	Findings
Nosrati et al. [26]	Intraoperative outcomes	Subsurface cues such as pulsation patterns, textures, and colours within the operative field and preoperative imaging	Machine learning	The framework has application in non- visible and partially occluded structures during PN
DiDio et al. [27]	Intraoperative outcomes	Preoperative CT Pyelography	Computer vision to create HA3D model	The model used to selectively apply pressure to the artery during PN
Klen et al. [28]	Postoperative outcomes	Data from 1099 operated RARC patients	Machine Learning	Identifying patients who are at high risk for complications after RC, and additional factors identified via ASA score, CPD, ACCI, and CHF
Checcucci et al. [29]	Postoperative outcomes	3D and non 3D prostate models from RARP and mpMRI	Multivariable linear regression models	No extracapsular extension in mpMRI and the use of 3D models during RARP lowered the incidence of positive margins

Abbreviations: PN = partial nephrectomy; CT = computed tomography; HA3D = hyper-accuracy 3D model; RC = radical cystectomy; RARC = robotic-assisted radical cystectomy; ASA = American Society of Anesthesiologists; CPD = chronic pulmonary disease; ACCI = age-adjusted Charlson comorbidity index; CHF = chronic heart failure; RARP = robotic-assisted radical prostatectomy; mpMRI = multiparametric MRI.

**Table 5 cancers-16-01775-t005:** Studies with postoperative applications.

Authors ^(Ref)^	Outcome	Data Source	Type of AI	Findings
Bagdadi et al. [30]	Intraoperative assessment	20 PLD videos	Machine learning	The machine learning model may accurately generate prostatectomy assessment and competence evaluation (PACE) scores
Haifler et al. [31]	Intraoperative assessment	Shortwave Raman spectroscopy data	Machine learning	Differentiation between malignant and benign renal tissue
Checcucci et al. [32]	Intraoperative assessment	Footage captured by the endoscope during RARP at 3-s intervals	Artificial neural network	Can predict bleeding
Porpiglia et al. [33,34]	Intraoperative assessment	HA3D models of the prostate from mpMRI used during RARP	Virtual augmented reality	Increased recognition of ECE
De Backer et al. [36]	Intraoperative assessment	Da Vinci recordings	Virtual augmented reality and deep learning	Detection of tools during renal surgery; motion and alignment artefacts during patient’s breathing
Amparore et al. [37]	Intraoperative assessment	HA3D model, NIRF firefly to enhance kidney visibility which was used as a reference point	Computer vision technology, Ignite for image anchoring	Overlapping the 3D model without manual assistance during AR-RAPN
Amparore et al. [38]	Intraoperative assessment	Prerecorded surgery videos	Convolutional neural network technology vs. indocyanine green injection	Overlapping the 3D model without manual assistance during RAPN even without indocyanine injection

Abbreviations: PLD = pelvic lymph node dissection; PACE = prostatectomy assessment and competence evaluation; RARP = robotic-assisted radical prostatectomy; HA3D = hyper-accuracy 3D; mpMRI = multiparametric MRI; ECE = extracapsular extension; AR-RAPN = augmented reality robotic-assisted partial nephrectomy; RAPN = robotic-assisted partial nephrectomy.

**Table 6 cancers-16-01775-t006:** Studies associated with postoperative outcomes prediction.

Authors ^(Ref)^	Outcome	Data Source	Type of AI	Findings
Hung et al. [40]	Postoperative outcomes	Automated measures of performance: overall duration, average period of inactivity, total length of all instruments’ paths, length of the path taken by the dominant/non-dominant instrument, length of the path taken, the time taken to move, the average velocity, and the amount of time spent idle for both the dominant and non-dominant instruments. adjustment of camera position, frequency, length of path, duration of movement, average velocity, duration of inactivity, energy consumption, exchange of a third arm utilisation of the clutch, exit the console	Three machine learning algorithms, using data from robot system and hospital length of stay	A machine learning algorithm designed to forecast the duration of hospitalisation and the duration of Foley catheter utilisation.
Hung et al. [41]	Postoperative outcomes	Automated measures of performance: Metrics pertaining to the measurement and analysis of time Measurements of the motion characteristics of an instrument, Metrics for measuring camera movement Metrics for measuring the articulation of the Endowrist Metrics for system events Characteristics of patients: age, body mass index (BMI), PSA, Gleason score prior to surgery, ASA classification, surgical duration, lymphadenectomy scope, urethropexy, nerve-sparing, median lobe, abnormal Gleason score, pathological staging size of the prostate gland, surgical margins, radiotherapy	DL model (DeepSurv)	A predictive model capable of determining continence outcomes following robotic radical prostatectomy; the emphasis lies on performance indicators rather than patient characteristics.

Abbreviations: PSA = prostate-specific antigen; ASA = American Society of Anesthesiologists.

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
