# Peer review of "Artificial Intelligence in Urologic Robotic Oncologic Surgery: A Narrative Review"

_cancers, 2024, doi:10.3390/cancers16091775_

Round 1
Reviewer 1 Report
Comments and Suggestions for Authors
I read the paper titled “Artificial Intelligence in Urologic Robotic Oncologic Surgery: A Narrative Review”.
The article is well written and I enjoyed reading it. This review provides a panoramical overview on the current applications of AI in urologic robotic oncologic surgery.
I would suggest the author few further articles to be discussed in their manuscript, considering their potential significant impact on clinical practice.
In the paragraph “Intraoperative Application of Artificial Intelligence in Surgical Procedures”, I would include the following articles:
- Amparore D, Checcucci E, Piazzolla P, Piramide F, De Cillis S, Piana A, Verri P, Manfredi M, Fiori C, Vezzetti E, Porpiglia F. Indocyanine Green Drives Computer Vision Based 3D Augmented Reality Robot Assisted Partial Nephrectomy: The Beginning of "Automatic" Overlapping Era. Urology. 2022 Jun;164:e312-e316. doi: 10.1016/j.urology.2021.10.053. Epub 2022 Jan 19. PMID: 35063460.
- Amparore D, Sica M, Verri P, Piramide F, Checcucci E, De Cillis S, Piana A, Campobasso D, Burgio M, Cisero E, Busacca G, Di Dio M, Piazzolla P, Fiori C, Porpiglia F. Computer Vision and Machine-Learning Techniques for Automatic 3D Virtual Images Overlapping During Augmented Reality Guided Robotic Partial Nephrectomy. Technol Cancer Res Treat. 2024 Jan-Dec;23:15330338241229368. doi: 10.1177/15330338241229368. PMID: 38374643; PMCID: PMC10878218.
In the paragraph “Addressing the Issue of Haptic Feedback” I would include the following article:
Piana A, Gallioli A, Amparore D, Diana P, Territo A, Campi R, Gaya JM, Guirado L, Checcucci E, Bellin A, Palou J, Serni S, Porpiglia F, Breda A. Three-dimensional Augmented Reality-guided Robotic-assisted Kidney Transplantation: Breaking the Limit of Atheromatic Plaques. Eur Urol. 2022 Oct;82(4):419-426. doi: 10.1016/j.eururo.2022.07.003. Epub 2022 Aug 16. PMID: 35985902.
In this article, the augmented reality was used to avoid iliac artery plaques during the arteriotomy in robotic-assisted kidney transplantation. In fact, there is no haptic feedback from the vessels when the robotic instruments are used. In the discussion, the authors hypothesize the use of artificial intelligence for an automatic 3DVM superimposition over the real vessel. But, a possible further application of AI in this field, would be to address the haptic feedback of atheromatic vessels. I think this point would be very interesting to be proposed.
Author Response
Dear Εditor and Reviewers,
We are grateful for your time and effort invested to review our manuscript and provide such valuable comments to improve its quality and readability for Journal’s audience. We have carefully gone through all of them and provide below a point-by-point answer. We remain at your disposal for anything further that may be needed.
With kindest regards,
Lazaros Tzelves
MD, MSc, PhD, FEBU, ECFMG
Ass. Professor in Urology, National and Kapodistrian University of Athens
Reviewer 1
1) ‘’I read the paper titled “Artificial Intelligence in Urologic Robotic Oncologic Surgery: A Narrative Review”. The article is well written and I enjoyed reading it. This review provides a panoramical overview on the current applications of AI in urologic robotic oncologic surgery.
I would suggest the author few further articles to be discussed in their manuscript, considering their potential significant impact on clinical practice.
In the paragraph “Intraoperative Application of Artificial Intelligence in Surgical Procedures”, I would include the following articles:
- Amparore D, Checcucci E, Piazzolla P, Piramide F, De Cillis S, Piana A, Verri P, Manfredi M, Fiori C, Vezzetti E, Porpiglia F. Indocyanine Green Drives Computer Vision Based 3D Augmented Reality Robot Assisted Partial Nephrectomy: The Beginning of "Automatic" Overlapping Era. Urology. 2022 Jun;164:e312-e316. doi: 10.1016/j.urology.2021.10.053. Epub 2022 Jan 19. PMID: 35063460.
- Amparore D, Sica M, Verri P, Piramide F, Checcucci E, De Cillis S, Piana A, Campobasso D, Burgio M, Cisero E, Busacca G, Di Dio M, Piazzolla P, Fiori C, Porpiglia F. Computer Vision and Machine-Learning Techniques for Automatic 3D Virtual Images Overlapping During Augmented Reality Guided Robotic Partial Nephrectomy. Technol Cancer Res Treat. 2024 Jan-Dec;23:15330338241229368. doi: 10.1177/15330338241229368. PMID: 38374643; PMCID: PMC10878218.
In the paragraph “Addressing the Issue of Haptic Feedback” I would include the following article:
Piana A, Gallioli A, Amparore D, Diana P, Territo A, Campi R, Gaya JM, Guirado L, Checcucci E, Bellin A, Palou J, Serni S, Porpiglia F, Breda A. Three-dimensional Augmented Reality-guided Robotic-assisted Kidney Transplantation: Breaking the Limit of Atheromatic Plaques. Eur Urol. 2022 Oct;82(4):419-426. doi: 10.1016/j.eururo.2022.07.003. Epub 2022 Aug 16. PMID: 35985902.
In this article, the augmented reality was used to avoid iliac artery plaques during the arteriotomy in robotic-assisted kidney transplantation. In fact, there is no haptic feedback from the vessels when the robotic instruments are used. In the discussion, the authors hypothesize the use of artificial intelligence for an automatic 3DVM superimposition over the real vessel. But, a possible further application of AI in this field, would be to address the haptic feedback of atheromatic vessels. I think this point would be very interesting to be proposed. ‘’
Answer
We wish to thank the Reviewer for the kind comments and valuable suggestions of further references to add. All proposed references are now added in the updated version.
Reviewer 2 Report
Comments and Suggestions for Authors
This review summarizes the application of artificial intelligence and machine learning in urologic robotic oncologic surgery. Overall, it covers some publications in this domain. It would be great if the selection criteria were clarified, as we do not know whether all publications were considered from the text. There are many redundancies in the main text, and therefore, it would be great to significantly reduce redundancy in the main text. Comments on the Quality of English Language
Minor editing is sophisticated.
Author Response
Dear Εditor and Reviewers,
We are grateful for your time and effort invested to review our manuscript and provide such valuable comments to improve its quality and readability for Journal’s audience. We have carefully gone through all of them and provide below a point-by-point answer. We remain at your disposal for anything further that may be needed.
With kindest regards,
Lazaros Tzelves
MD, MSc, PhD, FEBU, ECFMG
Ass. Professor in Urology, National and Kapodistrian University of Athens
Reviewer 2
1) ‘’This review summarizes the application of artificial intelligence and machine learning in urologic robotic oncologic surgery. Overall, it covers some publications in this domain. It would be great if the selection criteria were clarified, as we do not know whether all publications were considered from the text. ‘’
Answer
We wish to thank the Reviewer for the kind comments and valuable suggestion. According to methods section, all original articles published in English language and not performed in animals were included. This has been rephrased now in methods section to make it more clear to the reader.
2) ‘’ There are many redundancies in the main text, and therefore, it would be great to significantly reduce redundancy in the main text.’’
Answer
We wish to thank the Reviewer for the valuable suggestion. The whole text has been reviewed and any redundancies detected have been removed.
Reviewer 3 Report
Comments and Suggestions for Authors
I examined your study titled "Artificial Intelligence in urologic robotic oncologic surgery: A narrative review" in detail. I list the points that I see missing in the study in bullet points. First of all, why was the Simple Summary section written before the abstract section needed? While reading the article, I felt like I read the abstract twice. Especially the last sentences are repetitive. Machine learning is mentioned in almost the majority of the Introduction section. Such a subheading can be opened in the materials and methods section. The Introduction section should stick to its main purposes. At the end of this section, why the study was done, its contributions to the literature, etc. should be presented. This section should be concluded with a paragraph containing the organization of the article. When studies on the subject in the literature are examined, there are sentences such as "In a ground-breaking study, Fard et al. were pioneers in utilizing machine learning algorithms to evaluate the competence of robotic surgery abilities". This sentence is a very general sentence. Which machine learning methods were used and what are the advantages and disadvantages of this method. The articles reviewed in the summary should be examined in detail. These statements of mine are also valid for other studies. Which data sets were used in the presented studies and which results were obtained should be presented in a comparative table at the end of each section. I would also like to state that the similarity rate is high. Spelling and grammatical errors are at a high level in the study.
Comments on the Quality of English LanguageSpelling and grammatical errors are at a high level in the study.
Author Response
Dear Εditor and Reviewers,
We are grateful for your time and effort invested to review our manuscript and provide such valuable comments to improve its quality and readability for Journal’s audience. We have carefully gone through all of them and provide below a point-by-point answer. We remain at your disposal for anything further that may be needed.
With kindest regards,
Lazaros Tzelves
MD, MSc, PhD, FEBU, ECFMG
Ass. Professor in Urology, National and Kapodistrian University of Athens
Reviewer 3
1) ‘’I examined your study titled "Artificial Intelligence in urologic robotic oncologic surgery: A narrative review" in detail. I list the points that I see missing in the study in bullet points. First of all, why was the Simple Summary section written before the abstract section needed? While reading the article, I felt like I read the abstract twice. Especially the last sentences are repetitive.’’
Answer
We wish to thank the Reviewer for the effort made to review our manuscript and provide valuable comments. Regarding the Simple summary section, it is a requirement from Journal to provide a simple summary on top of the abstract, thus we have included this simple summary before the abstract section.
2) ‘’Machine learning is mentioned in almost the majority of the Introduction section. Such a subheading can be opened in the materials and methods section.’’
Answer
We wish to thank the Reviewer for this valuable suggestion. Machine learning subheading now has been created in the appropriate section.
3) ‘’The Introduction section should stick to its main purposes. At the end of this section, why the study was done, its contributions to the literature, etc. should be presented. This section should be concluded with a paragraph containing the organization of the article.’’
Answer
We wish to thank the Reviewer for this valuable suggestion. A paragraph now has been added to the end of introduction section, analyzing the purpose of the study, its contribution to the literature, while at the same time the organization of article was also added.
4) ‘’When studies on the subject in the literature are examined, there are sentences such as "In a ground-breaking study, Fard et al. were pioneers in utilizing machine learning algorithms to evaluate the competence of robotic surgery abilities". This sentence is a very general sentence. Which machine learning methods were used and what are the advantages and disadvantages of this method. The articles reviewed in the summary should be examined in detail. These statements of mine are also valid for other studies. Which data sets were used in the presented studies and which results were obtained should be presented in a comparative table at the end of each section.
Answer
We thank the Reviewer for the effort made to provide valid and valuable comments for our manuscript. Based on this suggestion, we have further analyzed studies and also tables 1-6 now are added at the end of each section as suggested by the Reviewer.
5) ‘’I would also like to state that the similarity rate is high. Spelling and grammatical errors are at a high level in the study.’’
Answer
We thank the Reviewer for this suggestion. The whole text has been reviewed again for spelling and grammatical error and appropriate changes were made.
Round 2
Reviewer 1 Report
Comments and Suggestions for Authors
Dear authors, thank you for having considered my suggestions.
Comments on the Quality of English LanguageMinor language editing
Author Response
Dear Editor and Reviewers,
We are grateful for your efforts in reviewing our manuscript. We have amended the manuscript according to your comments and this is now demonstrated in our revised manuscript. We remain at your disposal for anything further needed.
With kind regards,
Lazaros Tzelves
MD, MSc, PhD, FEBU, ECFMG
Ass. Professor in Urology,
National and Kapodistrian University of Athens
Reviewer 1
1) ‘’Minor language editing’’
Answer
We thank the Reviewer for giving us the opportunity to revise our manuscript. Language is now revised and edited accordingly.
Reviewer 3 Report
Comments and Suggestions for Authors
In your study titled "Artificial Intelligence in urologic robotic oncologic surgery: A narrative review", it is seen that serious changes were made after the revision. I congratulate the researchers on this. I have listed some of the missing points in the article. Examining the articles specified for the relevant section, where the machine learning section is very detailed. In the Machine Learning section, studies on the subject are mentioned. In the relevant section, you can review "https://www.mdpi.com/2075-4418/13/7/1299", "https://www.mdpi.com/2076-3417/13/17/9926 ". Your second part remained as one paragraph after revision. It would be more appropriate to place it under the third section in the relevant section. If a study can be found, it would be nice to expand Table 2, Table 3, and Table 6. Spelling and grammatical errors in the article should be reviewed. Why was Doi used in Figure 1? Can't a reference be made here too? 4) ‘’When studies on the subject in the literature are examined, there are sentences such as "In a ground-breaking study, Fard et al. were pioneers in utilizing machine learning algorithms to evaluate the competence of robotic surgery abilities". This sentence is very general. Which machine learning methods were used and what are the advantages and disadvantages of this method? These statements of mine are also valid for other studies. In the previous revision, the deficiencies I mentioned in number 4 were addressed. However, it has not been completely resolved. I believe that correcting the last deficiencies I mentioned will improve the quality of your article.
Comments on the Quality of English LanguageIt is important to review spelling and grammatical errors.
Author Response
Dear Editor and Reviewers,
We are grateful for your efforts in reviewing our manuscript. We have amended the manuscript according to your comments and this is now demonstrated in our revised manuscript. We remain at your disposal for anything further needed.
With kind regards,
Lazaros Tzelves
MD, MSc, PhD, FEBU, ECFMG
Ass. Professor in Urology,
National and Kapodistrian University of Athens
Reviewer 2
1) ‘’ In your study titled "Artificial Intelligence in urologic robotic oncologic surgery: A narrative review", it is seen that serious changes were made after the revision. I congratulate the researchers on this. I have listed some of the missing points in the article. Examining the articles specified for the relevant section, where the machine learning section is very detailed. In the Machine Learning section, studies on the subject are mentioned.’’
Answer
We thank the Reviewer for the kind words and effort made to review our manuscript.
2) ‘’In the relevant section, you can review "https://www.mdpi.com/2075-4418/13/7/1299", "https://www.mdpi.com/2076-3417/13/17/9926 ".
Answer
Thank you very much for this suggestion. Those studies are now added in the reference list.
3) ‘’Your second part remained as one paragraph after revision. It would be more appropriate to place it under the third section in the relevant section.’’
Answer
We thank the Reviewer for this suggestion. We regret to say that we kept the second part as one paragraph and didn’t place it under the third section in order not to change article structure, since it will affect the logical sequence for readers.
4) ‘’If a study can be found, it would be nice to expand Table 2, Table 3, and Table 6. Spelling and grammatical errors in the article should be reviewed.’’
Answer
We thank the Reviewer for this comment. We have extensively searched literature and no more relevant articles could be found, so those tables were not further expanded. We have also reviewed our manuscript and amended grammatical and spelling errors accordingly.
5) ‘’Why was Doi used in Figure 1? Can't a reference be made here too? ‘’
Answer
We thank the Reviewer for this suggestion. Reference instead of DOI is now used in Figure 1
6) ‘’When studies on the subject in the literature are examined, there are sentences such as "In a ground-breaking study, Fard et al. were pioneers in utilizing machine learning algorithms to evaluate the competence of robotic surgery abilities". This sentence is very general. Which machine learning methods were used and what are the advantages and disadvantages of this method? These statements of mine are also valid for other studies. In the previous revision, the deficiencies I mentioned in number 4 were addressed. However, it has not been completely resolved. I believe that correcting the last deficiencies I mentioned will improve the quality of your article.
Answer
We thank the Reviewer for this suggestion. Further changes were made according to your request.
Round 3
Reviewer 3 Report
Comments and Suggestions for Authors.